# Compartmentation of cGMP Signaling in Induced Pluripotent Stem Cell Derived Cardiomyocytes during Prolonged Culture

**DOI:** 10.3390/cells11203257

**Published:** 2022-10-17

**Authors:** Maria Faleeva, Ivan Diakonov, Prashant Srivastava, Masoud Ramuz, Gaia Calamera, Kjetil Wessel Andressen, Nadja Bork, Lorenza Tsansizi, Marie-Victoire Cosson, Andreia Sofia Bernardo, Viacheslav Nikolaev, Julia Gorelik

**Affiliations:** 1Cardiac Section, National Heart and Lung Institute (NHLI), Faculty of Medicine, Imperial College London, Hammersmith Campus, Du Cane Road, London W12 0NN, UK; 2Department of Pharmacology, Institute of Clinical Medicine, University of Oslo and Oslo University Hospital, P.O. Box 1057 Blindern, 0316 Oslo, Norway; 3German Center for Cardiovascular Research, University Medical Center Hamburg-Eppendorf and Institute of Experimental Cardiovascular Research, Martinistrasse 52, 20251 Hamburg, Germany; 4The Francis Crick Institute, 1 Midland Road, London NW1 1AT, UK

**Keywords:** cGMP, phosphodiesterase, stem cells, cardiomyocytes, FRET

## Abstract

The therapeutic benefit of stimulating the cGMP pathway as a form of treatment to combat heart failure, as well as other fibrotic pathologies, has become well established. However, the development and signal compartmentation of this crucial pathway has so far been overlooked. We studied how the three main cGMP pathways, namely, nitric oxide (NO)-cGMP, natriuretic peptide (NP)-cGMP, and β_3_-adrenoreceptor (AR)-cGMP, mature over time in culture during cardiomyocyte differentiation from human pluripotent stem cells (hPSC-CMs). After introducing a cGMP sensor for Förster Resonance Energy Transfer (FRET) microscopy, we used selective phosphodiesterase (PDE) inhibition to reveal cGMP signal compartmentation in hPSC-CMs at various times of culture. Methyl-β-cyclodextrin was employed to remove cholesterol and thus to destroy caveolae in these cells, where physical cGMP signaling compartmentalization is known to occur in adult cardiomyocytes. We identified PDE3 as regulator of both the NO-cGMP and NP-cGMP pathway in the early stages of culture. At the late stage, the role of the NO-cGMP pathway diminished, and it was predominantly regulated by PDE1, PDE2, and PDE5. The NP-cGMP pathway shows unrestricted locally and unregulated cGMP signaling. Lastly, we observed that maturation of the β_3_-AR-cGMP pathway in prolonged cultures of hPSC-CMs depends on the accumulation of caveolae. Overall, this study highlighted the importance of structural development for the necessary compartmentation of the cGMP pathway in maturing hPSC-CMs.

## 1. Introduction

Pathologies resulting in heart failure are frequently associated with aberrant cyclic adenosine monophosphate (cAMP) signaling, a secondary signaling molecule that promotes cardiomyocyte contractility [1]. In order to produce healthy cardiac muscle contraction, cAMP is regulated via interplay between the cAMP and cyclic guanosine monophosphate (cGMP) pathway [2]. Cyclic GMP is a secondary messenger that promotes cardiomyocyte relaxation [2,3]. Cyclic GMP is synthesized from guanosine triphosphate (GTP) by two forms of guanylyl cylcases. Soluble guanylyl cyclase (sGC) is stimulated by nitric oxide (NO) and forms the NO-cGMP pathway [4,5]. Cellular receptors NPR1 (GC-A) and NPR2 (GC-B) are stimulated by natriuretic peptides, atrial natriuretic peptide (ANP), brain natriuretic peptide (BNP), and C-type natriuretic peptide (CNP), the activators of particulate GC-A and GC-B, respectively, thus forming the NP-cGMP pathway [4,5]. The third cGMP pathway can be activated via adrenergic stimulation of the β_3_-Adrenoreceptor (AR) [5,6]. Its colocalization with endothelial nitric oxide synthase (eNOS) then further promotes NO synthesis in the NO-cGMP pathway [6,7].

The study of cGMP has been predominantly elusive due to its relatively low cellular concentrations as compared to cAMP. By using Förster Resonance Energy Transfer (FRET) microscopy it is possible to examine physiologically relevant cAMP and GMP localised in subcellular microdomains [8]. Compartmentation of cAMP signalling predominantly occurs through caveolae and transverse tubules (T-tubule) nanodomains [9]. By localising β_2_-adrenoreceptors (AR) and their associated signalling complexes, these functional signalosomes maintain a high level of specificity of cAMP signalling [9,10,11]. Likewise, cGMP is compartmentalised throughout the cell. Initial cGMP compartmentation is provided by the localisation of sGC and NPR1/2; sGC is diffusely localized throughout the cytosol whilst NRP1/2 are localized on the plasma membrane [12]. Further compartmentation of cGMP is achieved by its hydrolysis. Phosphodiesterases (PDE) are exclusively responsible for hydrolyzing the synthesized intracellular cGMP. Out of the seven PDEs that have been found to be expressed in the myocardium, PDE5 [12] and PDE9 [13] are specific for cGMP, whilst PDE1, PDE2, and PDE3 are dual-substrate specific for cGMP and cAMP [14]. PDE2 and PDE3 are crucial for maintaining cAMP/cGMP crosstalk, with higher levels of cGMP resulting in an increase and decrease in the cAMP hydrolysis rate, respectively [5]. In adult cardiomyocytes, PDE1 and PDE5 isoforms have been shown to modulate the sGC-cGMP pools [12], PDE2 and PDE9 isoforms to modulate the NP-cGMP [4,13], and PDE3 isoforms to regulate the cGMP pools found both at the membrane [15,16], and in the cytosol [17]. However, PDE3 isoforms have a higher affinity for cAMP rather than cGMP [18], and thus are not typically regarded as having strong cGMP modulatory abilities in adult cardiomyocytes.

Given cGMP’s ability to regulate cAMP levels, the cytoprotective capabilities elicited by cGMP on the cardiac system have been well studied. Therapeutics such as organic nitrates [19] and PDE5 inhibitors [20] have been used as vasodilators. A sGC stimulator, BAY102-1189 (Vericiguat), that works by sensitizing and increasing the activation of sGC in low NO environments, has been shown to be effective against artery and pulmonary hypertension [21,22]. Recent research has shown that on top of its vasodilatory properties, cGMP possesses anti-fibrotic mechanisms [21]. By inhibiting the transforming growth factor (TGF)-β pathway, increased cGMP levels can inhibit the transcription of extracellular matrix (ECM) genes such as collagen (COL)2A/3A [21,23], and thus their deposition in lung, renal, and cardiac fibrosis. Blocking the TGF-β pathway further allows cGMP to inhibit the phenotypic switch that occurs during fibrotic diseases, such as the transition from fibroblast to myofibroblast that occurs in hypertrophic scarring [24], and activation of dermal fibroblasts in dermal fibrosis [21].

Using differentiated human pluripotent stem derived cardiomyocytes (hPSC-CM) [25] provides a unique opportunity to study the development of target pathways [26]. Although fully differentiated hPSC-CMs have been described as having a gene signature closer to that of fetal rather than adult cardiac tissue [27,28], various cardiac features do mature to a certain extent with time in culture [29]. Importantly, it has been shown that aging cells in culture is critical for the development of structural compartmentation such as caveolae regions [10]. Here, we employed an extended culture cell model to study in-depth the development of the cGMP pathway. By applying high-throughput FRET microscopy [30] and using a highly specific and selective cGMP sensor ScGI (EC_50_ of 200nm) [31], we provide the first comprehensive analysis of the maturation of the cGMP pathway in prolonged culture of hPSC-CMs. We show that after being aged in culture for 90 days, hPSC-CMs develop their cGMP signaling specificity through the hydrolysis of cGMP via PDEs in the NO-cGMP pathway and the localization of the β_3_-AR into developed caveolae regions, thus compartmentalizing the β_3_-AR-cGMP pathway.

## 2. Materials and Methods

### 2.1. Reagents

Unless otherwise specified, all reagents were obtained from Sigma-Aldrich, St. Louis, MO, USA.

### 2.2. hPSC Differentiation

IMR90-1 (WiCell Research Institute Inc, Madison, WI, USA) (IMR) human induced pluripotent stem cells (IMR-hPSCs) and WA09 (H9) human embryonic pluripotent stem cells (H9-hPSCs) (WiCell Research Institute Inc, Madison, WI, USA) were grown on growth factor reduced Matrigel (1:100 dilution) and mTeSR1 media (StemCell Technologies, Cambridge, UK) until differentiation started. The previously outlined protocol [25] was implemented to achieve cardiomyocyte differentiation from the IMR-hPSC-CMs, which were grown in culture for 5, 30, and 90 days post differentiation. H9-hPSCs were differentiated using a left ventricle specific differentiation method (manuscript under review) for 20 days post differentiation. CDI-hPSC-CMs (iCell) were purchased from Fujifilm Cellular Dynamics Inc., Madison, WI, USA [32] and were cultured for 40 days as per manufacturers’ instructions.

### 2.3. Quantitative Polymerase Chain Reaction (Q-PCR)

Cells were subjected to lysis with RNAzol and RNA was extracted from the cells following the RNAzol^®^ RNA extraction method. RNA samples were converted to cDNA through reverse transcriptase polymerase chain reaction (RT-PCR) by following the Applied Biosystems Protocol (ThermoFisher Scientific, Waltham, MA, USA). Q-PCR was performed using iTaq Universal SYBR^®^ Green Supermix (Bio-Rad, Watford, UK). The sequences for the primers are indicated in Appendix A.

### 2.4. Western Blotting

Cells were homogenized in lysis buffer containing: 1 mmol/L EGTA, 10 mmol/L HEPES, 150 mmol/L NaCl, 300 mmol/L sucrose, 1% Triton-X, and phosphatase and protease inhibitors (Roche, Welwyn Garden City, UK). Protein concentration was determined using Pierce BCA protein assay kit (ThermoFisher Scientific, Waltham, MA, USA). Protein lysates were mixed with 4 × SDS sample buffer containing: 200 mmol/L Tris-HCl, 8% SDS, 40% Glycerol, 0.4% Bromophenol blue, and 40 mmol/L DTT, and boiled at 95 °C for 5 min. Proteins (20 µg of total protein per lane) were size separated by sodium dodecyl sulfate polyacrylamide gel electrophoresis (SDS-PAGE) and transferred onto nitrocellulose membranes. For immunodetection primary GAPDH (5G4, dilution 1:160,000, Bio Trend, Koln, Germany), PDE1C (generous gift from C. Yan, dilution 1:1000), PDE2A (PD2A-101AP, dilution 1:500 Fabgennix, Frisco, TX, USA), PDE3A (generous gift from C. Yan, dilution 1:1000), PDE4B (ab170939, dilution 1:2500, Abcam, Cambridge, UK), PDE4D (ab171750, dilution 1:2500, Abcam, Cambridge, UK), and PDE5A (generous gift from S. Rybalkin, dilution 1:100) were used.

### 2.5. Transmission Electron Microscopy

Samples were prepared for transmission electron microscopy (TEM) as described previously [10]. Cells were fixed with 2.5% glutaraldehyde in cacodylate buffer, post fixed with osmium tetroxide, and embedded in Araldite. Cut sections were stained with uranyl acetate and lead citrate for visualization with TEM. For each sample, 10 fields of view were recorded from 10 random cells on a section at 20,000× magnification. The number of caveolae per μm of membrane was manually scored.

### 2.6. FRET Imaging

At the required age, hPSC-CMs were seeded at a density of 15,000 cells per plated on Fibronectin-coated glass bottom MatTek dishes (MatTek Corporation, Ashland, MA, USA) In this study, hPSC-CMs were transfected with the ScGI cGMP sensor, allowing for easy validation and identification of specific cell transfection via the fluorescent green microscope channel [31]. A 24h incubation with Lipofectamine™ 3000 Reagent (ThermoFisher Scientific, Waltham, MA, USA) was used, and manufacturer instructions were followed. Cells were imaged between 48–72 h post transfection. Prior to imaging, both types of cells were washed three times with FRET buffer (144 mM NaCl, 10 nM HEPES, 1 mM MgCl_2_, 5 mM KCL, pH 7.2–7.3) to halt spontaneous cell beating. All drugs were purchased from Bio-Techne Ltd., Abingdon, UK, and were diluted using the FRET buffer. Stimulation of cGMP production was achieved with either GSNO (5 μM) or CNP (500 nM). To explore the β_3_-AR-NO-cGMP pathway, β_1_- and β_2_-AR were inhibited with 100 nmol/L CGP20712A and 50 nmol/L ICI118551, respectively, 20 min prior to FRET imaging. Cells were subsequentially treated with isoproterenol (ISO, 100 nM). PDE inhibitors were used as follows: EHNA (10 μM, IC50 0.8 μM), a PDE2 inhibitor, Cilostamide (10 μM, IC50 27 nM), a PDE3 inhibitor, Sildenafil (1 μM, IC50 3.5 nM), a PDE5 inhibitor, PF-04447943 (1 μM, IC50 8 nM). 3-isobutyl-1-methylxanthine (IBMX, 100 μM) was used as a non-specific PDE inhibitor and saturator. The applied concentrations of inhibitors were chosen to elicit a fully saturating effect. To observe the effects of caveolae on cGMP signaling, cells were treated with methyl-β-cyclodextrin (MβCD) that resulted in the abolishment of all caveolae structures [33]. In total, 1–2 mM of MβCD was dissolved in RB+ media and administered to the cells. Cells were incubated for 1 h and then immediately imaged, following the FRET protocol. A novel high throughput FRET system was used [30], which enabled the changes in cGMP to be recorded from numerous cells. Prior to commencing the experiment, the region of interest (ROI), i.e., the outline of the cell was defined, as well as the background. Photomicrographs were obtained every 6 s, and the background YFP/CFP values were subtracted from the YFP/CFP values emitted from the ROI. This allowed for live cell monitoring of YFP/CFP corrected ratio values vs. time. Model traces show the time taken for cGMP levels to change depending on the treatment, with time depicted on the x-axis, with the changing cGMP levels (YFP/CFP values) on the y-axis. Percentage of cGMP-FRET response for each condition was calculated from the initial baseline to the signal plateau post treatment. Calculations were made from 10 data points when plateau was achieved, during which there was minimal signal change.

### 2.7. RNA-SEQ Analysis

IMR-hPSC-CM cells were subjected to long-term maturation (30 and 90 days). The RNA from each time point were sequenced and mapped to the reference genome (hg38) using STAR version 2.5.4b [34]. The count data was normalised to log_2_ (count per million (CPM)). Further, to include the day five time-point, we used RNA-seq data published on the same cell lines for days five and 30. We performed a batch correction using the R package COMBAT [35].

H9-hPSC-CM cells were subjected to left ventricle cardiomyocyte differentiation. RNA from day 6, 20, and 40 samples was sequenced and mapped to the reference human genome assembly (hg38) using the publicly available nf-core rnseq pipeline v3.0 [36] with the STAR/RSEM [34,37] option. Ensembl release 95 transcript annotations were used to obtain gene-level abundance estimates. Heatmaps were generated using the normalised count data and presented as a mean of gene abundances across replicate groups. Data were additionally scaled per gene using a z-score to aid visualization. Rows (genes) were hierarchically clustered using a “complete” clustering method on a set of Euclidean distances. Sequencing data can be found in GSE203375 [38]. 

Single-cell RNA-seq data of adult cardiomyocytes were also obtained from published data with accession number GSE121893 [39]. Briefly, Wang et al. performed transcriptomic analysis of human hearts at single-cell resolution using the smart-seq 2 protocol from the cells from the hearts of normal individuals who died of non-cardiac illness. The cardiac cells were dispensed in 384 well plates, where RNAs were extracted for sequencing. The cells were later sequenced and annotated using known transcriptomic markers [39]. In this manuscript, we have extracted the cardiomyocyte cells from the two healthy hearts (N13: #cells = 270, N14: #cells = 245). The gene expression (gene counts) from cells was combined by “pseudo bulking”, i.e., by combining counts from all cardiomyocyte cells. The combined count data was used to calculate log_2_(CPM).

Spearman’s correlation based on fold changes was calculated to estimate the convergence of relevant pathways between hPSC-CMs and adult-CMs. The data is presented as a scatterplot using the ‘heatmap.2′ function from the gplots R package (version 3.1.1).

### 2.8. Statistical Analysis:

GraphPad Prism 8 software was utilized to conduct a statistical analysis on the results. To determine the effect of hPSC-CM ageing from day 30 to day 90 in gene expression analysis and FRET experiments, paired T-tests were corrected for multiple comparisons using the Holm–Sidak method. The data is expressed as the standard error of the mean (SEM) plus deviation. Significance was considered if the *p*-value was less than 0.05. *, **, ***, **** represent *p* < 0.05, *p* < 0.01, *p* < 0.001, and *p* < 0.0001, respectively.

## 3. Results

### 3.1. The cGMP-PKG Pathway Undergoes Maturation as hPSCs Are Cultured Up to Day 90

Initially, we analyzed the gene changes involved in the cGMP-PKG pathway in IMR-hPSC-CMs aged from day 5 to day 90 compared to that of adult cardiomyocytes (Figure 1a,b). A full list of the genes analyzed can be found in Appendix A. This was performed to determine whether increased culturing of hPSC-CMs would over time result in maturation of the cGMP pathway. As IMR-hPSC-CMs were cultured over time, their expression profile correlated more with adult cardiomyocytes. IMR-hPSC-CMs cultured for 30 and 90 days displayed a higher Spearman’s correlation of 0.71 and 0.72, respectively, whereas cells cultured for five days had a very poor Spearman’s correlation (0.267) (Figure 1b).

### 3.2. hPSC-CMs Modulate Their PDE Expression and Increase Their Structural Components throughout Ageing in Culture

Having validated that the cGMP pathway increases in maturation with prolonged culture of IMR-hPSC, we next investigated the transcript and protein expression of specific PDEs known to regulate the cGMP pathway. At the transcript level, PDE1C and PDE5A increased in expression throughout the prolonged culture of IMR-hPSC-CM (Figure 2a,b). However, their levels of protein expression were downregulated (PDE1C) or unchanged (PDE5A) when IMR-hPSC-CMs were kept in culture for 90 days (Figure 2c,d). PDE3A also exhibited a trend of upregulation at the transcript level but no change in its protein levels from day 30 to day 90. On the other hand, PDE2 was downregulated in older IMR-hPSC-CMs at both the transcript level (from D5 to D30/90) (Figure 2a,b) and at the protein level at day 90 (Figure 2c,d). We also observed downregulation at the transcript level for PDE9A and PDE1B. This data suggests that during development, cardiomyocytes regulate PDE expression at both the transcription and the translation level, as is the case for PDE1C. The expression of PDE isoforms was studied in another hPS-CM line (H9-derived) on both RNA and protein level (Appendix A) between day 20 and day 40 in culture. Similarly to the IMR-derived line, RNA-seq results show upregulation of PDE5A and PDE1C, whereas PDE3B, 9A, 1B, and 2A are downregulated. PDE3A has not changed. Selected PDE isoforms were checked for the expression of proteins on Western blots. Downregulation of PDE2A was confirmed, PDE3A has not changed, while PDE5A has increased in expression. Surprisingly, PDE1C showed downregulation, contrary to the RNA data. Similar behavior of PDE1C was seen in the IMR-derived hPS-CMs (Figure 2).

### 3.3. hPSC-CMs Increase Their Structural Components over Time in Culture

We next explored the gene changes associated with cGMP compartmentation (Figure 3a). Our RNA-SEQ dataset, validated via qPCR, revealed that IMR-hPSC-CMs do not significantly alter the gene expression of NRP1/2 throughout prolonged culture. However, after ageing for 90 days, there is an upregulation of the ADBR3 gene, which is responsible for encoding the β_3_-AR (Figure 3a,b). Upregulation of all three genes over time in culture was, however, detected in a different cell line, H9-hPSC-CMs (Appendix A), highlighting small differences across lines. Most structural genes are upregulated from Day 5 and maintain consistent expression from Day 30 in both cell lines (Figure 3a and Appendix A). However, the genes involved in caveolae development, CAV1/2/3, are upregulated further when hPSC-CMs are cultured for 90 days (Figure 3a and Appendix A).

The increase in caveolae over time in culture was further validated with transmission electron microscopy (TEM). Quantification of caveolae numbers for the IMR-hPSC-CMs aged day 30 and day 90 has been done previously [10] and revealed that cells at day 30 have 0.2 caveolae per micron, and at day 90 have 0.6 caveolae per micron. We also examined cardiomyocytes generated from another hPSC line CDI-hPSC-CMs. Notably, CDI-hPSC-CMs showed a vast number of caveolae structures (1.3 caveolae per micron) after being cultured for 40 days (Figure 3c–e). In keeping with previous work, we further showed that it is possible to diminish the number of caveolae at the cell membrane by treating hPSC-CMs with MCβD (Figure 3d,e). Together these data suggest that only little physical compartmentalization of cGMP signaling might be possible in early hPSC-CM cultures and that cell lines have different levels of caveolae development at different times in culture.

### 3.4. Levels of cGMP Synthesis Change Depending on the Pathway Stimulated and hPSC-CM Age

Knowing that prolonged culture of hPSC-CMs may contribute to changes in cGMP signaling, we next undertook FRET microscopy to examine the physiological changes in cGMP compartmentation (Figure 4a). The production of cGMP upon the stimulation of the three different pathways, NO-, NRP2-, and β_3_-AR-NOS, was analyzed in IMR-hPSC-CMs (Figure 4b–g). As cells aged in culture from day 30 to day 90, there was a significant decrease in the amount of cGMP produced upon the stimulation of the NO-cGMP pathway with GSNO but no changes were observed when the NRP2- and β_3_-AR-NOS pathways were stimulated (Figure 4b–d). Furthermore, the amount of cGMP produced upon the stimulation of the β_3_-AR pathway was remarkably lower than the amount produced upon the activation of the other two pathways (Figure 4d). Together this data shows that upon their stimulation, these three different pathways produce different levels of cGMP, which may, in part, be regulated via changing of PDE activity.

### 3.5. Compartmentation of the NO-cGMP Pathway Is Regulated via PDE Activity

We next analyzed the role of PDE1, PDE2, PDE3, PDE5, and PDE9 during the activation of the three cGMP pathways. We started by looking at the NO-cGMP pathway, which had demonstrated extensive signaling changes as the cells aged in culture. In day 30, IMR-hPSC-CMs inhibition of PDE1 and PDE3 resulted in the most noticeable increase in cGMP levels (Figure 5). As the cells aged to day 90, the level of recorded cGMP upon PDE3 inhibition did not increase as significantly as when the cells were younger. However, PDE1 retained its prominent role in cGMP hydrolysis, significantly regulating the NO-cGMP alongside PDE2 and PDE5. Due to PDE1, PDE2, and PDE5 showing the highest levels of cGMP hydrolysis in the NO-cGMP pathway, FRET imaging was conducted with simultaneous inhibition of all three PDEs (Figure 5g,h). This resulted in a maximal saturation response, validating that PDE1, PDE2, and PDE5 are the key enzymes hydrolyzing NO-cGMP. Inhibition of PDE9 resulted in minimal changes in cGMP levels throughout cell ageing in culture.

### 3.6. Maturation of the NP-cGMP Pathway Results in cGMP Signalling That Is Not Locally Restricted

The effect of the PDEs was then analyzed in the NRP2-NP-cGMP pathway. As with the NO-cGMP pathway, in day 30 IMR-hPSC-CMs, inhibition of PDE3 resulted in the most significant increase of cGMP production and its contribution to NP-cGMP hydrolysis decreased dramatically at day 90 (Figure 6a,d). The contribution of PDE1 for compartmentalization of this pathway was, on the other hand, minimal at both time points. In fact, at day 90, the individual inhibition of all PDEs resulted in a similar but minimal increase of cGMP levels (Figure 6a). To ascertain if physical compartmentalization played a role in the regulation of the NP-cGMP pathway, cells were treated with MβCD to induce caveolae depletion. No differences were observed in untreated and treated conditions in the amount of cGMP produced upon NPR2-cGMP pathway stimulation with CNP in IMR-hPSC-CMs aged day 30 or day 90 (Appendix A). Together, this suggested the NP-cGMP pathway is not regulated by either PDEs or the presence of caveolae.

### 3.7. Compartmentation and Development of β3-AR-cGMP Pathway Is Dependent on Caveolae Structure Formation

Finally, the regulatory effect of the PDEs in the β_3_-AR-NOS-cGMP pathway was explored. None of the PDE inhibitors elicited a large increases of cGMP production regardless of the time point analyzed (Figure 7). However, this was not in line with the aforementioned findings of this paper, which revealed that β_3_-AR expression levels increased as hPSC-CMs were aged in culture (Figure 3b).

To explore whether we were unable to detect a sizable production of cGMP via this pathway due to compartmentation of this receptor in nanodomains, we treated cells with MβCD to remove the caveolae (Figure 8). The experiment was performed on three different hPSC-CMs: IMR, H9, and CDI cells. MβCD treatment in day 30 IMR-hPSC-CMs produced no change of cGMP upon β_3_-AR pathway stimulation with ISO as compared to the untreated group (Figure 8a,b). However, day 90 IMR-hPSC-CMs, which have an increased number of caveolae structures, produced more cGMP in the MβCD treated group as opposed to the untreated group (Figure 8c,d). Likewise, aged CDI-hPSC-CMs, which have the highest number of caveolae structures observed out of the three cell lines at the time points tested, showed the highest increase of cGMP upon MβCD treatment (Figure 8i,j). H9-hPSC-CMs aged day 20 have little caveolae structures and showed minimal changes in cGMP levels upon MβCD treatment as compared to their controls (Figure 8e–h). Collectively, this data shows that longer culture periods are required for increasing the number of caveolae structures as well as for their maturation and use as signalosomes in the context of the β_3_-cGMP pathway.

## 4. Discussion

Numerous studies have highlighted the therapeutic benefit of harnessing cGMP for the treatment against heart failure [19,20] and fibrotic pathologies [21,24]. Thus, it is crucial to understand how the compartmentation and signal specificity of the cGMP pathway develops. Using hPSC-CM as a developmental cell model, we have explored how the three major signaling pathways of cGMP, namely, NO-cGMP, NP-cGMP, and β3-AR-cGMP, mature over time in culture.

We found that the expression profile of genes involved in the cGMP-PKG pathway in older hPSC-CM cultures correlated more strongly with adult cardiomyocytes than early-stage cultures. Regarding PDE expression, at the transcript level, PDE1C, PDE3A, and PDE5A were the most upregulated in older hPSC-CMs. However, we also found that PDE expression may be regulated in a complex way since PDEs’ transcript and protein expression did not necessarily match; for example, PDE1C and PDE3A transcript expression was upregulated from day 30 to day 90 but at the protein level their expression was either downregulated or maintained, respectively. This may be due to extensive post translational modifications or short half-lives of the PDE proteins. On the other hand, structural proteins such as caveolin show stable upregulation at the transcript level in later cultures of hPSC-CMs, and TEM validated the presence of more and better developed caveolin structures in older hPSC-CM cultures [10]. Possible differences in the genetic background could explain the differences seen in caveolae number/development within the different hPSC-CMs. These observations confirmed that indeed time in culture leads to changes in morphology and expression of cGPM pathway linked genes, which may reflect changes in the development of this pathway in vivo.

The NO-cGMP pathway is stimulated via sGC and occurs throughout the cytoplasm of the cell [12]. We saw that the NO pathway in IMR-hPSC-CMs is inhibited more by PDEs on day 90 than on day 30, demonstrating a higher level of compartmentalization and pathway maturity in hPSC-CMs that were cultured for longer. The maximum production of NO-cGMP is higher in day 90 cells rather than day 30, as can be seen in Figure 5a upon IBMX treatment. At day 30, the predominant regulators of NO-cGMP were found to be PDE1 and PDE3. However, at day 90, the role of PDE3 in the NO-cGMP pathway was diminished, with the regulation being maintained by PDE1, PDE2, and PDE5. This may signify a more mature phenotype as PDE1 and PDE5 have been established as key regulators of the cytosolic cGMP signaling pathways in adult cardiomyocytes [12,40]. Surprisingly, PDE2 has high activity in regulating the NO-cGMP pathway. PDE2 hydrolyzes both cAMP and cGMP, and its preference changes depending on the relative abundance of both molecules in the cell [41]. When the concentration of cGMP is high, the N-terminal GAF domain of PDE2 becomes occupied by cGMP. This causes an allosteric modification that lowers its Km for cAMP [42] and subsequently an increase of cAMP hydrolysis and a decrease in cGMP hydrolysis. PDE2 has thus been proposed as an antihypertrophic agent [43,44]. However, this hypothesis has been contested by others [45]. Our results show that PDE2 can increase its ability to hydrolyze cGMP in older hPSC-CM cultures, which makes it questionable if PDE2 is an antihypertrophic agent. However, we were unable to confirm this because no analyses of hypertrophy or fibrosis were conducted. Moreover, we cannot rule out that these changes could reflect the relative immature phenotype of the cells even in aged cardiomyocytes.

The NP-cGMP pathway is stimulated by the activation of NRP1/2 receptors on the cell membrane by ANP, BNP, or CNP [4,5]. In adult cardiomyocytes, the CNP-NP-cGMP pathway has been shown to produce a diffuse signal, without any individual PDE being responsible for its hydrolysis [46]. Therefore, we can once again confirm the maturity of our day 90 cGMP pathway. In day 30 hPSC-CMs, we found that PDE3 works as a regulator of cGMP hydrolysis after stimulation of the NP-pathway. However, aged hPSC-CMs don’t rely on any particular PDE to regulate NP-cGMP hydrolysis and are therefore more like mature adult cardiomyocytes. It is not surprising that PDE3 is found to be involved in both NP- and NO-cGMP pathways, as different isoforms of this enzyme can be found in the cytoplasm [21] or near the cell membrane [47,48]. PDE3 is known to have a double specificity. However, its Vmax for cAMP is approximately 10-fold higher than for cGMP, making it an important factor in cAMP hydrolysis [18]. We may therefore speculate, that the basal level of cAMP increases as hPSC-CMs age in culture, enabling PDE3 to regulate this signaling molecule at greater level, which is further demonstrative of a more mature cellular phenotype.

Another interesting PDE that has been recently identified to play a role in pathological phenotypes is PDE9 [5,13]. However, in our study we could not confirm whether PDE9 has a role in regulating the NO- or NP-cGMP pathways. We conclude that PDE9′s role in the cGMP pathways in mature cells under healthy physiological conditions is minimal.

In a mature β_3_-AR cGMP pathway, the β_3_-AR receptor is compartmentalized to the caveolae [6] and T-tubule [7] regions. Using three separate hPSC-CM models, IMR, H9, and CDI cells, we were further able to examine the changes in β_3_-AR-cGMP pathway physical compartmentation throughout development. We demonstrated that the proper compartmentation of the β_3_-AR is highly dependent on the initial development and maturation of caveolae on the hPSC-CM membrane. Cells with less developed caveolae, such as day 30 IMR-hPS-CMs and day 20 H9-hPSC-CMs were consistently unable to produce a strong cGMP signal upon stimulation of the β_3_-AR, despite expressing the receptor. However, in cells with more developed caveolae such as day 90 IMR-hPSC-CMs and CDI-hPSC-CMs, the signal derived from β_3_-AR stimulation increased significantly. We can thus conclude that caveolae maturation is essential for the compartmentalization of the β_3_-AR, and that time in culture is essential for the development of more and more mature caveolae. Moreover, this pathway, contrary to the NO-cGMP and NRP-cGMP pathways, relies on nanodomains (caveolae) to achieve signal specificity.

## 5. Conclusions

The balance of the various cGMP pathways’ activity shifts during cardiomyocyte development and is more tightly regulated in more mature cardiomyocyte cultures. The NO-cGMP pathway is heavily regulated by PDEs and despite some discordance between PDEs’ expression at the transcript and protein level, we found a significant change in their activity throughout the maturation of this pathway (Figure 9). This demonstrates that a given PDE expression is tightly regulated and that its activity is not only based on expression, but it is likely to depend on its localization within cardiomyocytes. Interestingly, while PDE3 appears to regulate the NP-cGMP pathway earlier in development, neither PDEs nor physical compartmentalization appears to play a role in the compartmentalization of this pathway in older hPSC-CM cultures. The β_3_-AR-cGMP pathway, on the other hand, becomes segregated in caveolae in older hPSC-CMs exhibiting a more mature cGMP-PKG signaling pathway.

## Figures and Tables

**Figure 1 cells-11-03257-f001:**
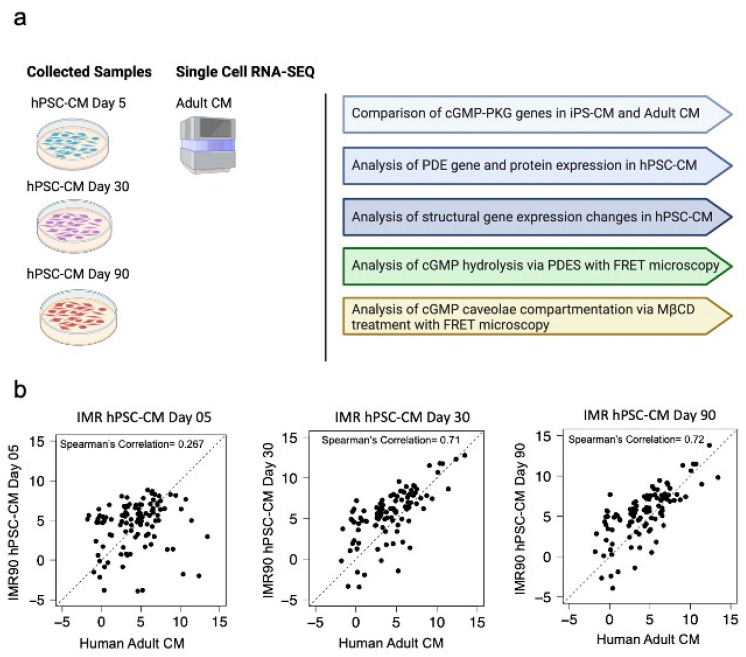
(**a**) Schematic of the executed experiments for the study. (**b**) Spearman’s correlation in fold changes of cGMP-PKG pathway gene expression in IMR-hPSC-CMs cultured for 5, 30, and 90 days as compared to adult cardiomyocytes.

**Figure 2 cells-11-03257-f002:**
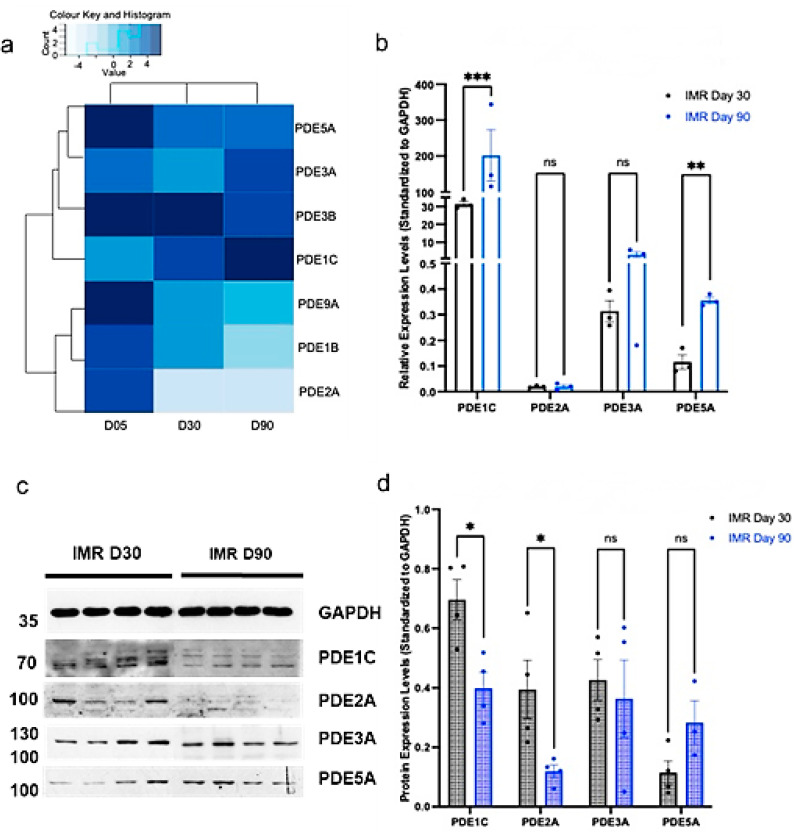
Phosphodiesterase (PDE) gene and protein expression in IMR90 induced pluripotent stem cell cardiomyocytes (IMR-hPSC-CM) aged day 5, 30, and 90. (**a**) Heatmap representing the expression of cGMP-hydrolysing PDEs from the RNA-SEQ dataset in IMR-hPSC-CMs aged day 5, 30, and 90. (**b**) Gene expression of PDEs in IMR-hPSC-CMs cultured for 30 and 90 days. Western blot depiction (**c**) and protein expression quantification (**d**) of PDEs in IMR-hPSC-CMs cultured for 30 and 90. * *p* < 0.1; ** *p* < 0.01; *** *p* < 0.001.

**Figure 3 cells-11-03257-f003:**
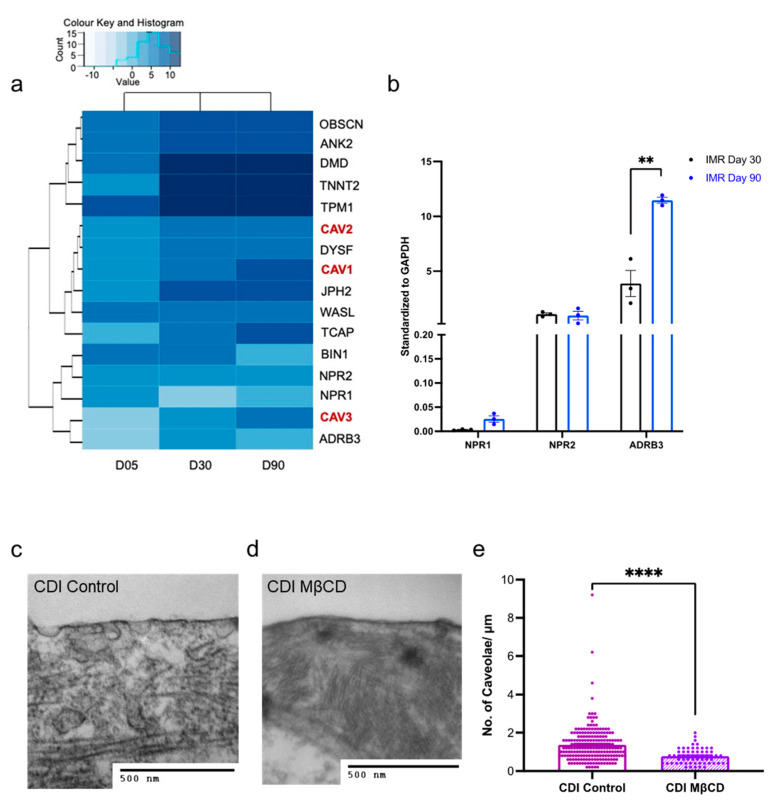
Expression of structural components in hPSC-CMs aged day 5, 30, and 90. (**a**) Heatmap of structural gene expression in IMR-hPSC-CMs aged day 5, 30, and 90. (**b**) Gene expression of Natriuretic peptide receptors (NPR) 1/2 and ADBR3 in IMR-hPSC-CMs aged day 30 and 90. Representative transmission electron microscopy (TEM) images of control (**c**) and methyl-β-cyclodextrin (MβCD) (**d**) treated CDI-hPSC-CMs. (**e**) Quantitation of caveolae per micrometre in CDI-hPSC-CMs. ** *p* < 0.01; **** *p* < 0.0001.

**Figure 4 cells-11-03257-f004:**
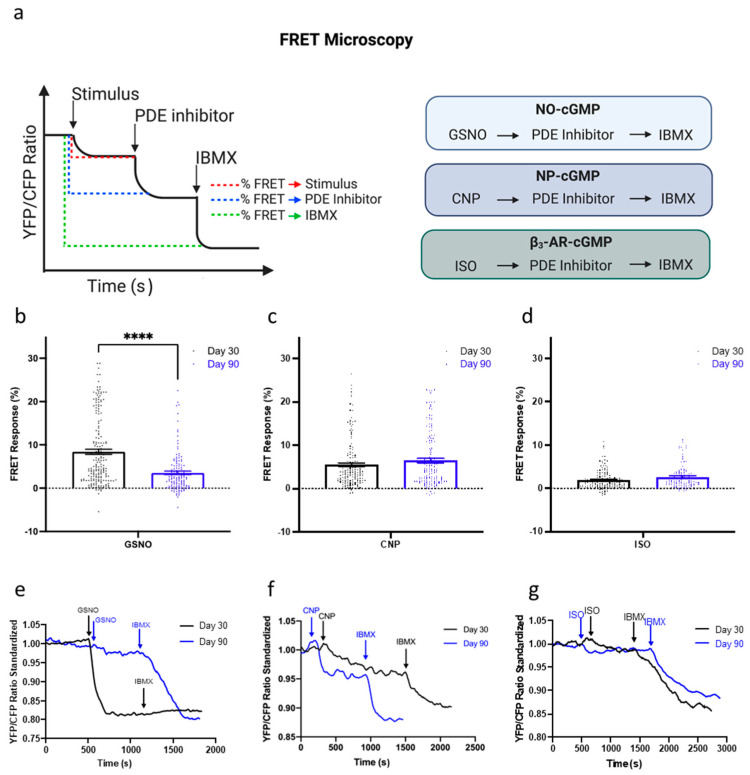
FRET microscopy for the analysis of the cGMP pathways. (**a**) Schematic of FRET microscopy experimental design. The production of cGMP upon the stimulation of three different pathways; nitic oxide (NO)- (**b**,**e**), natriuretic peptide receptor (NRP)2 (**c**,**f**) and β_3_-adrenoreceptor (AR) (**d**,**g**), via S-Nitrosoglutathione (GSNO), C-type natriuretic peptide (CNP), and Isoproterenol (ISO), respectively, with the corresponding traces shown below. All data is composed of three independent experimental batches. **** *p* < 0.0001.

**Figure 5 cells-11-03257-f005:**
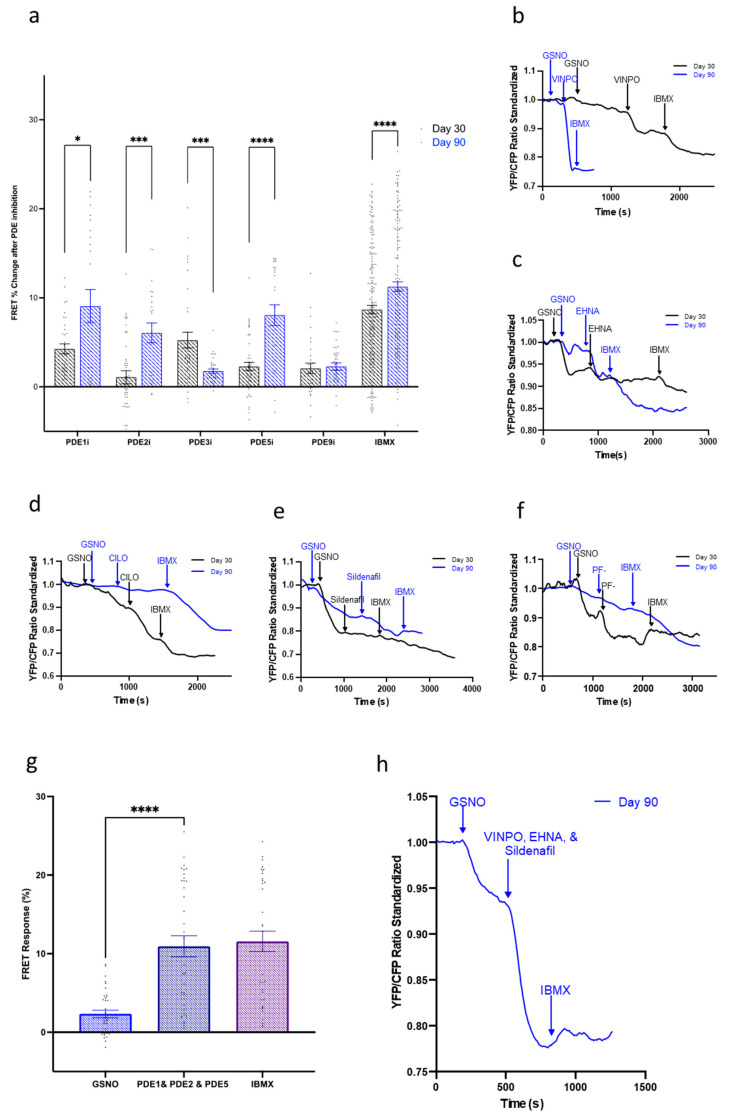
(**a**) Regulation of the nitric oxide (NO)-cGMP pathway by PDEs during aging of IMR-hPSC-CMs from day 30 to day 90. Model traces of cGMP production after stimulation via S-Nitrosoglutathione (GSNO) and subsequent (**b**) PDE1 inhibition via vinpocetine (VIMPO), (**c**) PDE2 inhibition via EHNA, (**d**) PDE3 inhibition via Cilostamine (CILO), (**e**) PDE5 inhibition via Sildenafil, and (**f**) PDE9 inhibition via PF-04447943 (PF-), and saturation via 3-isobutyl-1-methylxanthine (IBMX). (**g**,**h**) Bar chart (**g**) and model trace (**h**) illustrating the effect of simultaneous PDE1, PDE2, and PDE5 inhibition after NO-cGMP pathway stimulation via GSNO. All data is composed of three independent experimental batches. * *p* < 0.1; *** *p* < 0.001; **** *p* < 0.0001.

**Figure 6 cells-11-03257-f006:**
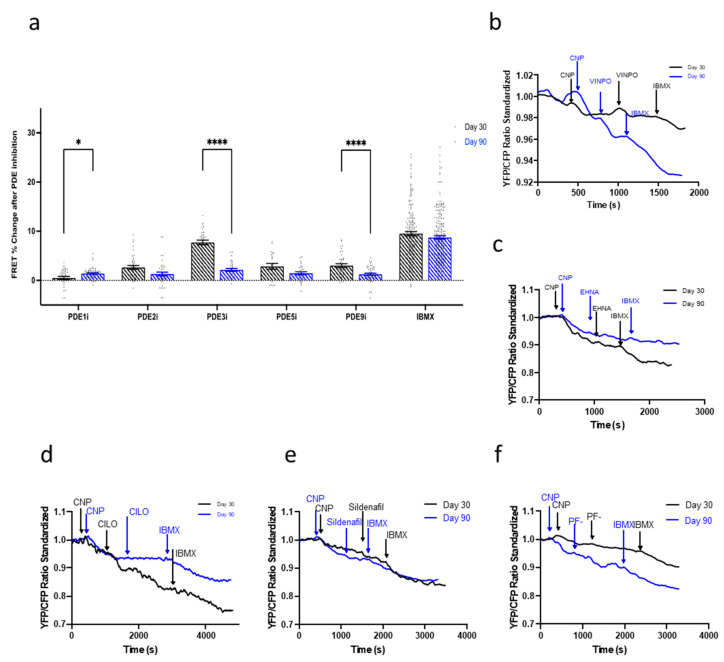
(**a**) Regulation of the natriuretic peptide receptor (NPR2)-C-type natriuretic peptide (CNP)-cGMP pathway by PDEs throughout the aging of IMR-hPSC-CMs in culture from day 30 to day 90. Model traces of cGMP production after the situation via CNP and subsequent (**b**) PDE1 inhibition via vinpocetine (VIMPO), (**c**) PDE2 inhibition via EHNA, (**d**) PDE3 inhibition via Cilostamine (CILO), (**e**) PDE5 inhibition via Sildenafil, and (**f**) PDE9 inhibition via PF-04447943 (PF-), and saturation via 3-isobutyl-1-methylxanthine (IBMX). All data is composed of three independent experimental batches. * *p* < 0.1; **** *p* < 0.0001.

**Figure 7 cells-11-03257-f007:**
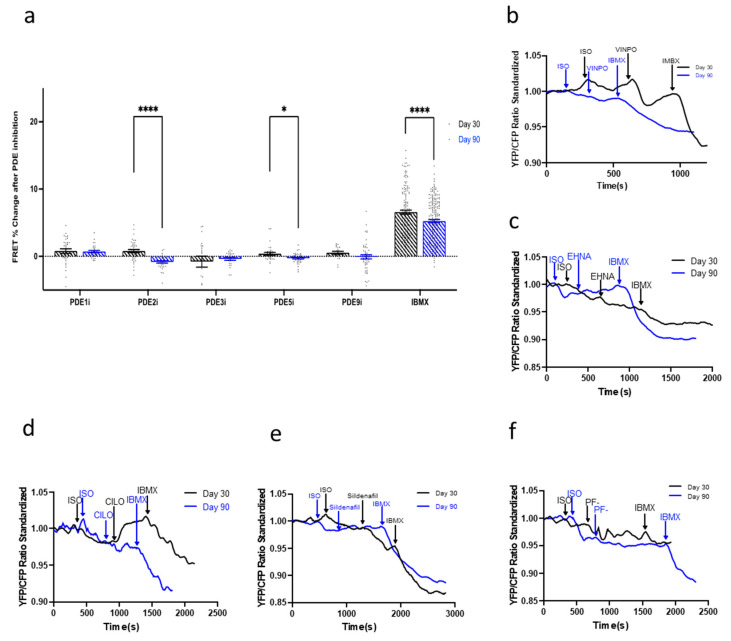
(**a**) The regulation of the β_3_-adrenoreceptor (AR)-cGMP pathway by PDEs throughout the aging of hiPSC-CMs in culture from day 30 to day 90. Model traces of cGMP production after the inhibition of β_1_- and β_2_-AR and situation of β_3_ via isoproterenol (ISO) and subsequent (**b**) PDE1 inhibition via vinpocetine (VIMPO), (**c**) PDE2 inhibition via EHNA, (**d**) PDE3 inhibition via Cilostamine (CILO), (**e**) PDE5 inhibition via Sildenafil, and (**f**) PDE9 inhibition via PF-04447943 (PF-), and saturation via 3-isobutyl-1-methylxanthine (IBMX). All data is composed of three independent experimental batches. * *p* < 0.1; **** *p* < 0.0001.

**Figure 8 cells-11-03257-f008:**
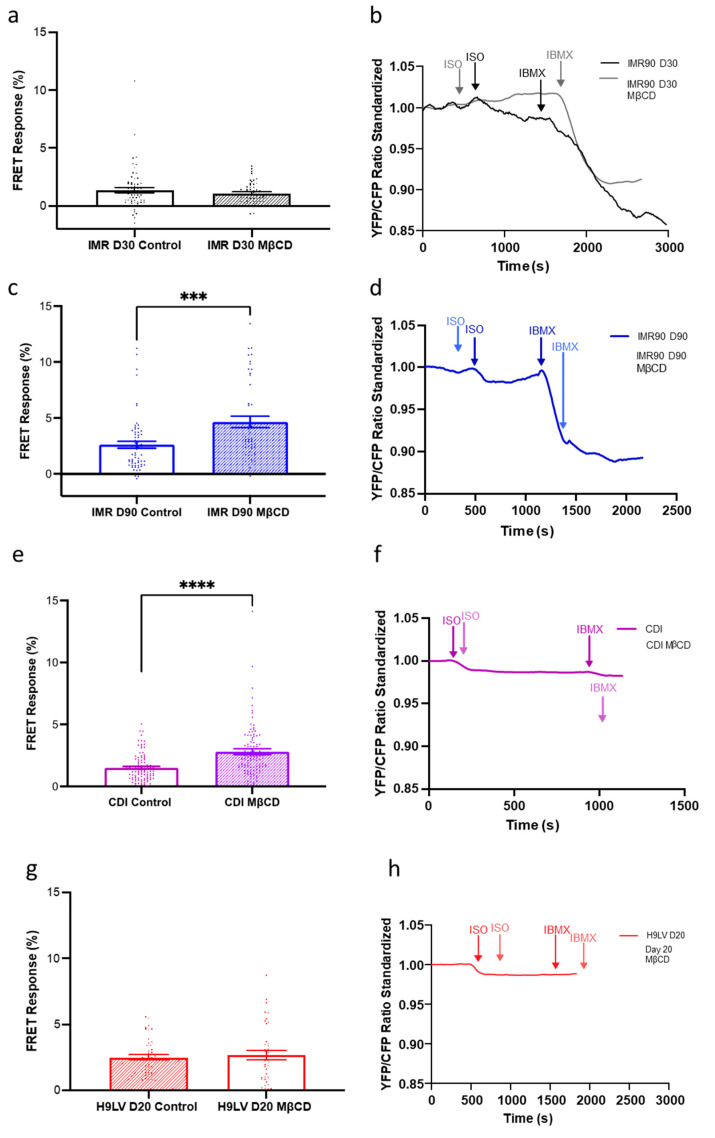
The effect of methyl-β-cyclodextrin (MβCD) treatment on the β_3_-adrenoreceptor (AR)-cGMP pathway in different lines of hPSC-CMs. Bar chart and model trace illustrating the effect of MβCD treatment on the cGMP production upon β_3_ -AR stimulation with isoproterenol (ISO) in (**a**,**b**) IMR-hPSC-CMs (IMR) Day 30, (**c**,**d**) IMR-hPSC-CMs (IMR) Day 90, (**e**,**f**) CDI-hPSC-CMs (CDI) grown in our lab for 40 days, (**g**,**h**) H9-hPSC-CMs (H9LV) Day 20. All data is composed of three independent experimental batches. *** *p* < 0.001; **** *p* < 0.0001.

**Figure 9 cells-11-03257-f009:**
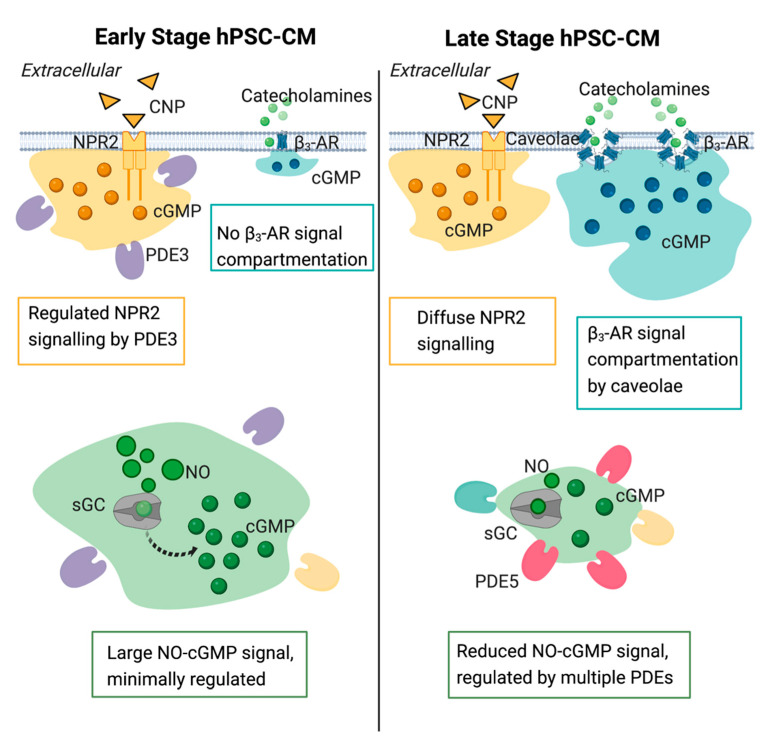
Schematic of the three cGMP pathways via nitric oxide (NO), natriuretic peptide receptor (NPR), and β_3_-adrenoreceptor (AR) stimulation in induced pluripotent stem cell derived cardiomyocytes.

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
