# Peer review of "Compartmentation of cGMP Signaling in Induced Pluripotent Stem Cell Derived Cardiomyocytes during Prolonged Culture"

_cells, 2022, doi:10.3390/cells11203257_

Round 1
Reviewer 1 Report
This paper proposed by Maria Faleeva et al, entitled “Compartmentation of cGMP Signaling in Induced Pluripotent Stem Cell Derived Cardiomyocytes During Prolonged Culture”. The authors studies how three cGMP pathways in human induced pluripotent stem cell derived cardiomyocytes mature over time in culture. This study is certainly important and of timely concern. However, several issues must be addressed before it can be accepted for publication.
1. In the methodology section the authors have not included how they transfected hiPSC-CM.
1. The hiPSC-CMs are non-dividing cells, and the efficiency of transfection is important to monitor and to achieve best efficiency of transfection specific time points after seeding are required. How many days did they transfected the cells prior to experiment and how they confirmed the cells were transfected with ScGIcGMP?
2. What is the seeding density of hiPSC-CMs?
3. In the line 142 and 143 they have mentioned “samples were prepared for TEM as described previously. They have cited reference (16) “Kenan Y, Murata T, Shakur Y, Degerman E, Manganiello VC. Functions of the N-514 terminal region of cyclic nucleotide phosphodiesterase 3 (PDE 3) isoforms. J Biol Chem. 515 2000, 275, 12331–8.” Doesn’t contain any information about TEM. Please elaborate how the TEM samples were prepared?
4. Please elaborate the single Cell RNA sequencing procedure. How the samples were prepared, quality control and details steps of analysis.
5. In the introduction authors mentioned that PDE5 and PDE 9 are specific for cGMP. Did they measure the expression of PDE9 at protein level? From the heat map data shown in Figure 2a, the PDE9 expression is upregulated at D05, as the cells mature it is downregulated (D90). According to authors discussion in the manuscript the PDE proteins have short half-lives or post-translational modifications may lead to increase or decrease in the PDE expression. It would be interesting to see the PDE9 expression at protein level.
6. In the line 241 the reference cited doesn’t match the context. Here the author talks about Quantification of caveolae numbers for the IMR-hPSC-CMs aged day 30 and day 90. They have cited reference (16) “Kenan Y, Murata T, Shakur Y, Degerman E, Manganiello VC. Functions of the N-514 terminal region of cyclic nucleotide phosphodiesterase 3 (PDE 3) isoforms. J Biol Chem. 515 2000, 275, 12331–8.”
7. In the line 325 the authors mentioned previous findings but not cited any references in the text.
Minor corrections –
1) The whole manuscript needs to be edited to a consistent writing. It is hard to read and follow. Please, make appropriate changes.
2) Please follow consistency while abbreviating Human Induced Pluripotent Stem Cell Derived Cardiomyocytes- hiPSC-CMs.
Author Response
- In the methodology section the authors have not included how they transfected hiPSC-CM.
- The hiPSC-CMs are non-dividing cells, and the efficiency of transfection is important to monitor and to achieve best efficiency of transfection specific time points after seeding are required. How many days did they transfected the cells prior to experiment and how they confirmed the cells were transfected with ScGIcGMP?
We thank the reviewer for raising this. Further information regarding the transfection has now been added to explain better how the experiments were performed (in the Method section). Note that all experiments were done on transiently transfected cells (analysed 48-72 post transfection) and that indeed the transfection efficiency is not large but, given that we can see which cells have been transfected (they fluoresce in the green channel), we are not worried about the inefficiency of the protocol as we only analyse cells that have been successfully transfected.
- What is the seeding density of hiPSC-CMs?
We have now updated the manuscript to include this information; cells were seeded at 15,000 per MatTek plate. (text added in the Methods section).
- In the line 142 and 143 they have mentioned “samples were prepared for TEM as described previously. They have cited reference (16) “Kenan Y, Murata T, Shakur Y, Degerman E, Manganiello VC. Functions of the N-514 terminal region of cyclic nucleotide phosphodiesterase 3 (PDE 3) isoforms. J Biol Chem. 515 2000, 275, 12331–8.” Doesn’t contain any information about TEM. Please elaborate how the TEM samples were prepared?
We apologise for this mistake, we have now fixed the citation mistake and replaced it with the correct citation: “Hasan A, Mohammadi N, Nawaz A, Kodagoda T, Diakonov I, Harding SE, et al. Age-Dependent Maturation of iPSC-CMs Leads to the Enhanced Compartmentation of β2AR-cAMP Signalling. Cells 2020, 9, 2275.”
- Please elaborate the single Cell RNA sequencing procedure. How the samples were prepared, quality control and details steps of analysis.
The single cell RNA sequencing data used in this manuscript is not our own data set but instead part of the following manuscript: Wang L, Yu P, Zhou B, Song J, Li Z, Zhang M, Guo G, Wang Y, Chen X, Han L, Hu S. Single-cell reconstruction of the adult human heart during heart failure and recovery reveals the cellular landscape underlying cardiac function. Nat Cell Biol, 2020, 22, 108–119. doi: 10.1038/s41556-019-0446-7. We have expanded the methods in this section to explain better what we did with the data and how it was originally obtained by the authors that first published this dataset.
- In the introduction authors mentioned that PDE5 and PDE 9 are specific for cGMP. Did they measure the expression of PDE9 at protein level? From the heat map data shown in Figure 2a, the PDE9 expression is upregulated at D05, as the cells mature it is downregulated (D90). According to authors discussion in the manuscript the PDE proteins have short half-lives or post-translational modifications may lead to increase or decrease in the PDE expression. It would be interesting to see the PDE9 expression at protein level.
In this study we were limited in our protein investigation of PDE9 expression due to the antibodies available. While we still intend to investigate further the role of PDE9 in the future, in this manuscript we already present the RNA expression data for this gene as well as its contribution for cGMP signalling via out FRET analysis. Given that the data suggest little contribution of PDE9 for hPSC-CMs, it may be more interesting to study this gene in the context of pathology rather than in normal developing cells.
- In the line 241 the reference cited doesn’t match the context. Here the author talks about Quantification of caveolae numbers for the IMR-hPSC-CMs aged day 30 and day 90. They have cited reference (16) “Kenan Y, Murata T, Shakur Y, Degerman E, Manganiello VC. Functions of the N-514 terminal region of cyclic nucleotide phosphodiesterase 3 (PDE 3) isoforms. J Biol Chem. 515 2000, 275, 12331–8.”
We apologise for the mistake, the citation was fixed and now refers to the correct publication: “Hasan A, Mohammadi N, Nawaz A, Kodagoda T, Diakonov I, Harding SE, et al. Age-Dependent Maturation of iPSC-CMs Leads to the Enhanced Compartmentation of β2AR-cAMP Signalling. Cells 2020, 9, 2275.”
- In the line 325 the authors mentioned previous findings but not cited any references in the text.
The findings referred to data within this manuscript, i.e. to the identification of an upregulation of beta3-AR in aged hPSC-CMs as was discussed in Figure 3. We have revised the text to refer back to this figure (Page XXX, line XXX).
Minor corrections –
- The whole manuscript needs to be edited to a consistent writing. It is hard to read and follow. Please, make appropriate changes.
In keeping with the reviewer’s request, the manuscript has been re-edited and we have revised several parts to aid readability.
2) Please follow consistency while abbreviating Human Induced Pluripotent Stem Cell Derived Cardiomyocytes- hiPSC-CMs.
We are aware that not all the manuscript was consistent regarding the abbreviations but we have now consistently revised the text and figures. We would like to highlight that we did not use the term hiPSC-CMs because not all cell lines are induced pluripotent stem cells, the H9 line is an embryonic pluripotent stem cell line. As such, we have adopted the term human pluripotent stem cells as it includes both induced- and embryonic-derived pluripotent stem cells. We have however, explained in the methods what each individual line is.
Reviewer 2 Report
By applying high-throughput FRET microscopy and using a highly specific and selective cGMP sensor ScGI (EC50 of 200nm), the authors showed the comprehensive analysis of the maturation of the cGMP pathway in prolonged culture of hPSC-CMs. After being aged in culture for 90 days, hPSC-CMs developed their cGMP signaling specificity through the hydrolysis of cGMP via PDEs and localization of the β3-AR into developed caveolae regions.
Depending on human iPSCs, their characteristics are greatly different depending on the cell lines. The data shown in Fig. 2 is only taken from IMR90-derived cardiomyocytes. The authors cannot get data tendency or conclusions from only one cell line results. The authors should show the results of H9(ESCs)-derived cardiomyocytes results in Supplementary data and discuss whether the tendency are the same to IMR90-derived cardiomyocytes. Typically three cell lines are necessary to convince the data in general. CDI- hiPSC-CMs’s data are also necessary.
Which cell line data for Fig. 3a and 3b? if only one cell line results, this is unacceptable. At least three cell lines (H9(ESCs)-derived cardiomyocytes, IMR90-derived cardiomyocytes, and CDI- hiPSC-CMs) should be used for the experiments.
Which cell line data for Fig. 4? if only one cell line results, this is unacceptable. At least three cell lines (H9(ESCs)-derived cardiomyocytes, IMR90-derived cardiomyocytes, and CDI- hiPSC-CMs) should be used for the experiments.
Which cell line data for Fig. 5? if only one cell line results, this is unacceptable. At least three cell lines (H9(ESCs)-derived cardiomyocytes, IMR90-derived cardiomyocytes, and CDI- hiPSC-CMs) should be used for the experiments.
Which cell line data for Fig. 6? if only one cell line results, this is unacceptable. At least three cell lines (H9(ESCs)-derived cardiomyocytes, IMR90-derived cardiomyocytes, and CDI- hiPSC-CMs) should be used for the experiments.
Which cell line data for Fig. 7? if only one cell line results, this is unacceptable. At least three cell lines (H9(ESCs)-derived cardiomyocytes, IMR90-derived cardiomyocytes, and CDI- hiPSC-CMs) should be used for the experiments.
Which cell line data for Fig. 8? if only one cell line results (it seems one cell line results using IMR90-derived cardiomyocytes), this is unacceptable. At least three cell lines (H9(ESCs)-derived cardiomyocytes, IMR90-derived cardiomyocytes, and CDI- hiPSC-CMs) should be used for the experiments.
Author Response
Depending on human iPSCs, their characteristics are greatly different depending on the cell lines. The data shown in Fig. 2 is only taken from IMR90-derived cardiomyocytes. The authors cannot get data tendency or conclusions from only one cell line results. The authors should show the results of H9(ESCs)-derived cardiomyocytes results in Supplementary data and discuss whether the tendency are the same to IMR90-derived cardiomyocytes. Typically three cell lines are necessary to convince the data in general. CDI- hiPSC-CMs’s data are also necessary.
We agree with the reviewer in that studying multiple hPSC lines leads to more robust data and therefore the conclusions become more general. Unfortunately, due to constrain in funding and time constrains, we cannot provide a comprehensive analysis of all the parameters studied in this work. In particular extra experiments with FRET will be impossible to conduct at this stage. However, we are now providing in the supplementary file two extra figures showing extra data on the transcript and protein expression of PDE isoforms and selected relevant genes in the H9-derived hPSC-CMs. Reassuringly, the data generally fit with the results obtained for the IMR-derived hPSC-CMs.
Which cell line data for Fig. 3a and 3b? if only one cell line results, this is unacceptable. At least three cell lines (H9(ESCs)-derived cardiomyocytes, IMR90-derived cardiomyocytes, and CDI- hiPSC-CMs) should be used for the experiments.
We apologise for not having been clearer about which line we were presenting the data in this part of the figure. The data presented in figure 3a and 3b is for IMR-hPSC-CMs. In keeping with the reviewers’ request we have also now analysed the expression of these genes in a separate line, H9-hPSC-CMs, and this data is presented in Figure S3. Reassuringly, the gene expression trend is similar.
Which cell line data for Fig. 4? if only one cell line results, this is unacceptable. At least three cell lines (H9(ESCs)-derived cardiomyocytes, IMR90-derived cardiomyocytes, and CDI- hiPSC-CMs) should be used for the experiments.
We apologise for not having been clearer about which line the data from this figure corresponds to. The data presented in figure 4 is for IMR-hPSC-CMs. Sadly, due to financial and time constraints we were not able to repeat the experiments in a separate line.
Which cell line data for Fig. 5? if only one cell line results, this is unacceptable. At least three cell lines (H9(ESCs)-derived cardiomyocytes, IMR90-derived cardiomyocytes, and CDI- hiPSC-CMs) should be used for the experiments.
We apologise for not having been clearer about which line the data from this figure corresponds to. The data presented in figure 5 is for IMR-hPSC-CMs. Sadly, due to financial and time constraints we were not able to repeat the experiments in a separate line.
Which cell line data for Fig. 6? if only one cell line results, this is unacceptable. At least three cell lines (H9(ESCs)-derived cardiomyocytes, IMR90-derived cardiomyocytes, and CDI- hiPSC-CMs) should be used for the experiments.
We apologise for not having been clearer about which line the data from this figure corresponds to. The data presented in figure 6 is for IMR-hPSC-CMs. Sadly, due to financial and time constraints we were not able to repeat the experiments in a separate line.
Which cell line data for Fig. 7? if only one cell line results, this is unacceptable. At least three cell lines (H9(ESCs)-derived cardiomyocytes, IMR90-derived cardiomyocytes, and CDI- hiPSC-CMs) should be used for the experiments.
We apologise for not having been clearer about which line the data from this figure corresponds to. The data presented in figure 7 is for IMR-hPSC-CMs. Sadly, due to financial and time constraints we were not able to repeat the experiments in a separate line.
Which cell line data for Fig. 8? if only one cell line results (it seems one cell line results using IMR90-derived cardiomyocytes), this is unacceptable. At least three cell lines (H9(ESCs)-derived cardiomyocytes, IMR90-derived cardiomyocytes, and CDI- hiPSC-CMs) should be used for the experiments.
We apologise for not having been clearer about which lines the data from this figure correspond to. The data presented in figure 8 comes from 3 different lines: 8a-d data is for IMR-hPSC-CMs; e-f data is for CDI-hPSC-CMs; g-h data is for H9-hPSC-CMs.
Reviewer 3 Report
Major comments
1. One major issue in the manuscript is control data or validation of the FRET sensor. Given that these results underly the overall manuscript’s hypothesis, this data needs to be strengthened. FRET data in control undifferentiated cells, unstimulated cells and a negative binding partner should be included.
2. What are the concentrations of NO donor, ISO and CNP used? Were the cells titrated to determine the best concentration for physiologically relevant activation of the pathway. The concentration and incubation time of these pathway activators will have profound effects on results and interpretation of the data.
3. The inhibitors to each of the PDE are not totally specific. Authors should provide additional siRNA data atleast in the final PDE1/2/5.
4. The FRET quantification and analysis is not clear as currently described in the methods. Authors state they perform recordings every 6 seconds. How does this fit in with time taken for the inhibitors or stimulators to act and stabilize. Also how are the representative curves generated as shown in the figures, and what are the data points statistics are calculated from?
5. While the inhibition of caveoli formation is an interesting concept, data should be provided regarding health of the cell over the course of the experiment. Did it trigger other stress responses, cell death?
6. It is not clear how authors distinguish cAMP from cGMP signaling after inhibiting caveoli formation at 90d? Perhaps commercially available kits can be used to specifically measure cGMP levels?
Minor comment
1. The introduction discussion sections need a concise short paragraph summarizing the central questions and findings respectively. These sections are written as a narrative and the main message of the paper is very unclear.
Author Response
- One major issue in the manuscript is control data or validation of the FRET sensor. Given that these results underly the overall manuscript’s hypothesis, this data needs to be strengthened. FRET data in control undifferentiated cells, unstimulated cells and a negative binding partner should be included.
At the time of the first submission we were unable to properly reference this sensor as we were waiting for its characterization to be submitted. Thankfully, the manuscript has now been submitted to biorxiv: Gaia Calamera, Bernadin Ndongson-Dongmo, Dulasi Arunthavarajah, Mette Ovesen, Choel Kim, Finn Olav Levy, Kjetil Wessel Andressen, Lise Román Moltzau Natriuretic peptides protect against apoptosis and increase cGMP around cardiomyocyte mitochondria. bioRxiv 2022.08.22.504735; doi: https://doi.org/10.1101/2022.08.22.504735
- What are the concentrations of NO donor, ISO and CNP used? Were the cells titrated to determine the best concentration for physiologically relevant activation of the pathway. The concentration and incubation time of these pathway activators will have profound effects on results and interpretation of the data.
The reviewer raises an important point and one that we have taken in consideration by using them at a concentration higher than the IC50 of the drugs and for a period of time in keeping with the described effect of the drugs. The concentrations of GSNO, CNP, and ISO used in the study have now been reported appropriately in the methods and we report also the IC50 for these drugs (page XXX and lines xxx). Note that all inhibitors were titrated in house to determine their IC50 and ascertain the appropriate concentration at which they should be used in order to elicit a fully saturating effect (data not shown).
- The inhibitors to each of the PDE are not totally specific. Authors should provide additional siRNA data atleast in the final PDE1/2/5.
We agree with the reviewer that providing the results of siRNA experiments would strengthen the message of the present study. However due to logistic constrains we are unable to conduct such experiments at present. In future we are going to perform these experiments.
- The FRET quantification and analysis is not clear as currently described in the methods. Authors state they perform recordings every 6 seconds. How does this fit in with time taken for the inhibitors or stimulators to act and stabilize. Also how are the representative curves generated as shown in the figures, and what are the data points statistics are calculated from?
We added a more detailed description of how we calculate FRET response in the relevant paragraph in the Methods section. We rely on the published data on the doses of all pharmacological compounds we use in our experiments. Typically, we use the reagent above the respective EC50 (in many cases well above) and we assume we are safe to expect the effect of these drugs within minutes. We have 5-10 minutes before we take measurements. As our software allows for the immediate display of the FRET ration only while we are doing the experiments, we can control the time we allow for the treatment to run until a steady plateau is reached.
- While the inhibition of caveoli formation is an interesting concept, data should be provided regarding health of the cell over the course of the experiment. Did it trigger other stress responses, cell death?
The removal of cholesterol leading to the destruction of caveolae by methyl-cyclodextrin has been used many times before and indeed it has been validated in numerous publications (1: Agarwal SR, MacDougall DA, Tyser R, Pugh SD, Calaghan SC, Harvey RD. Effects of cholesterol depletion on compartmentalized cAMP responses in adult cardiac myocytes. J Mol Cell Cardiol. 2011; 50(3):500-9. doi:10.1016/j.yjmcc.2010.11.015. 2: Calaghan S, Kozera L, White E. Compartmentalisation of cAMP-dependent signalling by caveolae in the adult cardiac myocyte. J Mol Cell Cardiol. 2008; 45(1):88-92. doi: 10.1016/j.yjmcc.2008.04.004. This idea was not pioneered by our lab but it has been adopted successfully by us in another publication, for which reason we did not re-do any extra validations on the method.
- It is not clear how authors distinguish cAMP from cGMP signaling after inhibiting caveoli formation at 90d? Perhaps commercially available kits can be used to specifically measure cGMP levels?
We apologise if this was not clear before. The sensor used in this study is a highly selective and specific cGMP sensor with an EC50 of 200nm. cAMP cannot be detected by this sensor and was not examined in this study at any point. We refer back to the publication demonstrating the specificity of this sensor: Gaia Calamera, Bernadin Ndongson-Dongmo, Dulasi Arunthavarajah, Mette Ovesen, Choel Kim, Finn Olav Levy, Kjetil Wessel Andressen, Lise Román Moltzau Natriuretic peptides protect against apoptosis and increase cGMP around cardiomyocyte mitochondria. bioRxiv 2022.08.22.504735; doi: https://doi.org/10.1101/2022.08.22.504735
Minor comment
- The introduction discussion sections need a concise short paragraph summarizing the central questions and findings respectively. These sections are written as a narrative and the main message of the paper is very unclear.
The last paragraph of the introduction reads: “By applying high-throughput FRET microscopy [30] and using a highly specific and selective cGMP sensor ScGI (EC50 of 200nm) [31] we provide the first comprehensive analysis of the maturation of the cGMP pathway in prolonged culture of hPSC-CMs. We show that after being aged in culture for 90 days, hPSC-CMs develop their cGMP signaling specificity through the hydrolysis of cGMP via PDEs in the NO-cGMP pathway and the localization of the β3-AR into developed caveolae regions, thus compartmentalizing the β3-AR-cGMP pathway.”
We think this gives a thorough view of the aims and main conclusions, whereas Discussion section contains Conclusion part which provides the summary requested by the reviewer.
Round 2
Reviewer 2 Report
No further comments
Reviewer 3 Report
The comments have been addressed and the manuscript is overall improved.